# Molecular Epidemiology, Clinical Course, and Implementation of Specific Hygiene Measures in Hospitalised Patients with *Clostridioides difficile* Infection in Brandenburg, Germany

**DOI:** 10.3390/microorganisms11010044

**Published:** 2022-12-22

**Authors:** Esther E. Dirks, Jasminka A. Luković, Heidrun Peltroche-Llacsahuanga, Anke Herrmann, Alexander Mellmann, Mardjan Arvand

**Affiliations:** 1Unit for Hospital Hygiene, Infection Prevention and Control, Department of Infectious Diseases, Robert Koch Institute, 13353 Berlin, Germany; 2Institute for Microbiology and Hospital Hygiene, Carl-Thiem-Hospital, 03048 Cottbus, Germany; 3Institute of Hygiene, University Hospital Muenster and National Reference Center for Clostridioides Difficile, Münster Branch, 48149 Münster, Germany

**Keywords:** *Clostridioides difficile*, healthcare-associated infection, mortality, recurrence, hygiene measures, PCR-ribotyping, whole genome sequencing, cgMLST

## Abstract

(1) Background: *Clostridioides difficile* infections (CDI) have increased worldwide, and the disease is one of the most common healthcare-associated infections (HAI). This study aimed to evaluate the molecular epidemiology of *C. difficile*, the clinical outcome, and the time of initiation of specific hygiene measures in patients with CDI in a large tertiary-care hospital in Brandenburg. (2) Methods: Faecal samples and data from hospitalised patients diagnosed with CDI were analysed from October 2016 to October 2017. The pathogens were isolated, identified as toxigenic *C. difficile*, and subsequently subtyped using PCR ribotyping and whole genome sequencing (WGS). Data regarding specific hygiene measures for handling CDI patients were collected. (3) Results: 92.1% of cases could be classified as healthcare-associated (HA)-CDI. The recurrence rate within 30 and 90 days after CDI diagnosis was 15.7% and 18.6%, and the mortality rate was 21.4% and 41.4%, respectively. The most frequent ribotypes (RT) were RT027 (31.3%), RT014 (18.2%), and RT005 (14.1%). Analysis of WGS data using cgMLST showed that all RT027 isolates were closely related; they were assigned to two subclusters. Single-room isolation or barrier measures were implemented in 95.7% patients. (4) Conclusions: These data show that RT027 is regionally predominant, thus highlighting the importance of specific hygiene measures to prevent and control CDI and the need to improve molecular surveillance of *C. difficile* at the local and national level.

## 1. Introduction

*Clostridioides difficile* (formerly *Clostridium difficile; C. difficile*) is an anaerobic, endospore-forming, gram-positive, rod-shaped bacterium widely distributed in the environment and the intestines of humans and animals. It is a genetically diverse species, and a number of strains can produce toxins that might lead to intestinal inflammation [1,2]. The clinical spectrum of *C. difficile* infections (CDI) ranges from mild diarrhoea to pseudomembranous colitis, sepsis, and even death [3]. Currently, *C. difficile* is considered the most common cause of antibiotic-associated colitis and healthcare-associated diarrhoea in developed countries [4]. Little is known about the situation in developing countries, but the estimated prevalence is concerningly high [5]. Mortality rate and recurrence rate have been determined in a number of studies with varying outcomes. The direct mortality rate is between 1–5%, and recurrence ranges between 1% and 35% [6,7,8,9]. However, few studies followed patients up for more than 30 days.

Transmission of *C. difficile* is possible via oral ingestion of the pathogen or its spores, which can be transmitted directly or indirectly to other individuals [2]. As the spores of this bacterium remain in the environment for months or years and can show high tolerance to various cleaning agents and disinfectants, timely implementation of appropriate hygiene measures is imperative to prevent further spread [10].

From 2000 onward, an increase in CDI and severe courses was observed in Canada and North America, which correlated temporally with the emergence of a new epidemic strain (ribotype) that was referred to as hypervirulent and typed using PCR ribotyping as RT027 [11,12]. Shortly after, this strain was also observed in Europe [13]. Even though infection rates have improved in some countries in recent years, possibly in part due to the introduction of antibiotic stewardship, CDI still remains one of the most common healthcare-associated infections (HAIs) [6,14].

On the European level, member states and their hospitals can participate in a CDI surveillance that registers HA-CDI using a common European protocol. Surveillance using this protocol can be performed on different levels (light, medium, and advanced). Germany has been taking part in this model since 2021 [15].

Currently, on a national level, two forms of surveillance are in place in Germany. One is the mandatory notification of severe cases of CDI. Another is a voluntary hospital-based program called CDAD-KISS. Like the European protocol, this program registers HA-CDI cases. However, unlike the European protocol, the CDAD-KISS protocol only defines a case as an HA-CDI that can be linked to the same hospital [16]. Therefore, information regarding the rate of overall HA-CDI is missing.

None of these systems provides molecular surveillance, which makes them not suited to gaining data on the distribution of different strains, especially hypervirulent ones.

This study aimed to evaluate the molecular epidemiology of *C. difficile*, the clinical course of the disease (including a follow-up of patients up to 90 days), and the time of initiation of specific hygiene measures in patients with CDI in a hospital in Brandenburg, a federal state located in eastern Germany. In addition, we assessed the genetic relationship among the RT027 isolates collected in this study utilising whole-genome sequencing (WGS) and subsequent core genome Multilocus Sequence Typing (cgMLST), which has a higher discriminatory power than ribotyping, allowing for the analysis of clusters and outbreaks [17].

## 2. Materials and Methods

### 2.1. Study Setting and Sample Collection

This study was conducted with a prospective study design.

The study hospital provides specialised care with approximately 1200 beds and is the largest hospital in its state in terms of the number of beds. More than 100,000 patients are treated there annually as inpatients and/or outpatients. A total of 52,411 inpatients were treated in the year 2017.

All patients aged ≥18 years who were hospitalised in the study hospital between October 2016 and October 2017 and had laboratory-confirmed CDI were included in this study. Patients were excluded if they had already been enrolled in the study in the previous 90 days. One faecal sample per included patient was collected and stored for culture and further analysis as described below.

Information regarding age, sex, time of admission, duration of admission, and primary and secondary diagnosis was collected. In addition, a survey was conducted 30 and 90 days after the diagnosis regarding recurrence and mortality rates. This survey was performed regardless of whether the patient had already left the hospital. Data were collected only after written informed consent was obtained from the patients (or their legal guardians). In the informed consent form, the patient agreed to be contacted by telephone 30 days after the onset of symptoms of CDI. Furthermore, they agreed that their primary care physician might also be interviewed on this topic 90 days after the onset of symptoms of CDI. This procedure was approved by an ethics vote obtained from the Brandenburg State Medical Association.

Recurrence of CDI was defined as the recurrence of symptoms with positive laboratory detection after cessation of therapy and two to eight weeks after the last positive specimen [15]. The assignment of the infection as HA or community-associated (CA) was performed following the ECDC definitions [15]. Data regarding hygiene measures were collected using a questionnaire handed to the treating physician.

### 2.2. Laboratory Procedures

For the laboratory confirmation of CDI, samples were first tested for the group antigen glutamic acid dehydrogenase (GDH). All GDH-positive samples were tested with the same method for the presence of *C. difficile* toxins A and B (Liaison XL, DiaSorin, Saluggio, Italy). All GDH-positive materials were cultured, and suspect *C. difficile* colonies grown on a selective agar were subsequently identified by matrix-assisted laser desorption ionisation with time-of-flight analysis (Maldi-TOF) (Vitek MS, bioMérieux, Nürtigen, Germany) [18]. Next, confirmed *C. difficile* isolates were tested for toxin A and B production by rapid immunochromatographic assay (ImmunoCard Toxins A and B, Meridian, Bioscience, Cincinnati, OH, USA). The toxin-producing isolates were ribotyped based on the protocol used by Arvand et al. [19]. Briefly, DNA was extracted, and PCR was performed. Electrophoresis was conducted in a 1.8% agarose gel for 240 min at 85 Volt, and the bands were visualised under UV light. Evaluation was performed by comparing the band pattern with the pattern of known reference strains. Finally, as recently described [17], cgMLST was performed on all identified RT027 isolates using an Illumina MiSeq instrument (Illumina Inc., San Diego, CA, USA).

### 2.3. Statistical Analysis

The collected data were analysed with the program SPSS 21 for Windows (IBM, Armonk, NY, USA). Within the statistical analysis framework, absolute and relative frequencies, the mean value, the median, and the minimum and maximum values were calculated for various parameters. The normal distribution of the data was checked with the Kolmogorov-Smirnov test. Comparison between different patient subgroups was performed by means of the chi-square test. All *p*-values were reported in terms of exact 2-sided significance and to the 95% confidence interval.

## 3. Results

In total, 112 patients with laboratory-confirmed CDI met the inclusion criteria and were enrolled in this study. From these patients, 99 toxigenic *C. difficile* isolates were available for further analysis. The patient’s biological sex was male in 49.1% and female in 50.9%. The median age of the patients was 77.6 years (range 21–95 years). The median time from admission to onset of CDI was 9 days (range 0–94 days). Onset of CDI was defined here as the date of first sample collection. The prevalence of CDI was 0.28 per 100 Patients, and the incidence density was 0.35 per 1000 patient days. In total, 12.5% of CDI cases were classified as a severe course of disease.

One-hundred-two of 112 confirmed CDI cases could be classified as either HA-CDI, CA-CDI, or unclear infection according to the ECDC 2017 definition [15]. Ninety-four of 102 (92.1%) patients with available information were categorised as HA-CDI. Among these, most cases were linked to the study hospital (89/94; 94.7%), while a small proportion (5/94; 5.3%) had been associated with a stay in a healthcare facility other than the study hospital.

Patient follow-up was possible at 30 and 90 days after laboratory confirmation of CDI in 70 (62.5%) patients. In detail, 52/112 patients or their authorised relatives had agreed to be contacted for follow-up, and 18/112 patients remained hospitalised (5 patients) or died (13 patients) in the hospital during the follow-up period. Recurrence of CDI was observed in 11 (15.7%) and 13 (18.6%) patients within 30 and 90 days of diagnosis, respectively. These recurrences occurred after a median of 23 days (range 14–38). Mortality within 30 and 90 days of diagnosis of CDI was assessed in 70 patients, as described above. Fifteen (21.4%) and 29 (41.4%) patients died within 30 days and 90 days of diagnosis, respectively. Mean and median mortality rates and the minimum and maximum time to death for 30- and 90-day follow-ups are shown in Table 1. The Pearson chi-square test showed no statistically significant difference in overall mortality within 90 days of diagnosis between patients with *C. difficile* RT027 infection and patients with infection with other ribotypes (χ2 = 1.204; *p* = 0.273).

PCR ribotyping was performed for all 99 isolates that were available for further characterisation. In total, the isolates were assigned to 21 different RTs. RT027 was the most prevalent ribotype, accounting for 31 (31.3%) of isolates. Other prevalent RTs were 014 (n = 18; 18.2%), 005 (n = 14; 14.1%), 002 (n = 7; 7.1%), 023 (n = 4; 4.0%), and 078 (n = 3; 3.0%) (Figure 1). The remaining isolates (n = 22; 22.2%) belonged to other RTs that were sporadically encountered and in two isolates, the RT could not be determined.

The results of cgMLST typing of the 31 RT027 isolates from the study hospital are shown in Figure 2, along with 3 isolates that were collected in the same period of time from another hospital in the same region. Differences detected among the isolates ranged from 0–20 alleles. Using the cluster threshold of ≤6 alleles distance according to Bletz et al. [17], all isolates formed a single genotypic cluster that could be further separated in two subclusters comprising 6 (subcluster B) or 25 (subcluster A) isolates per cluster (Figure 2). Whereas the genotypes of two of the five CA-CDI isolates were marginal to the subcluster A (isolates Cd 90 and Cd 92), the remaining three CA-CDI isolates (Cd 68, Cd 107, and Cd 113) could not be separated from the centre of the subcluster A. Similarly, the genotypes of the three isolates from a neighbouring hospital (Cd 11, Cd 28, and Cd 37) clustered with all other isolates of the subcluster A. Further analysis of the epidemiological data did not reveal any association between a subcluster and a distinct ward or department.

The initiation of and compliance with hygiene measures, as stated in the infection prevention and control (IPC) guidance of the hospital, were assessed in 70 (62.5%) patients with CDI. Altogether, single-room isolation or barrier measures were implemented in 67 (95.7%) patients. In detail, barrier-nursing was performed for 37 (52.9%) patients, and 27 patients (40.0%) were placed in a single room. Two patients (2.9%) had already been in single-room isolation for other reasons. The time from onset of diarrhoea to implementation of isolation/barrier-nursing measures was, on average, 0.8 (±1.2) days. Transition to sporicidal surface disinfectants occurred in 47 (42.0%) patients with a microbiologically confirmed diagnosis of CDI. The time from laboratory telephone notification to switch of the surface disinfectant averaged 0.8 (±1.2) days.

## 4. Discussion

The presented study gives insight into CDI occurrence, the clinical course and outcome of the disease, circulating RTs, and the implementation of hygiene measures in patients diagnosed with CDI in a tertiary-care hospital in Brandenburg. Follow-up of the patients involves giving a valuable assessment of CDI recurrence and mortality rates. This information is rarely available for CDI cases in Germany since the follow-up after the patient’s discharge is very time consuming and rather complicated due to data-protection regulations.

Compared with the data of the National Reference Centre for Surveillance of Nosocomial Infections, the CDAD-KISS module, which were collected in 535 hospitals in Germany, the overall prevalence and incidence density of CDI were lower in our study (prevalence of 0.28 in this study versus 0.39 per 100 patients in KISS; incidence density 0.35 in this study versus 0.56 per 1000 patient-days in KISS) [20]. International data are hard to compare due to differences in study design and settings, but a meta-analysis published in 2019 showed that CDI’s overall prevalence and incidence density ranged from 0 to 3.51 cases per 100 patients and from 0.01 to 5 per 1000 patient days in individual studies conducted worldwide from 2005 to 2015 [1].

The most frequently identified RTs in the current study were RTs 027, 014, and 005 (Figure 1). Previous studies reported RT001 as the RT with the highest prevalence in Germany [21,22]. However, in the present study, this RT was identified in only 2.0% of patients with CDI. Furthermore, in a recent study from the national reference centre for *C. difficile*, RT001 was the second most abundant ribotype, followed by RT078. Interestingly, in the latter study, RT027 was found only occasionally [23]. By contrast, RT027 was identified as the most common RT in our study. Previous studies from Germany have reported a lower prevalence of *C. difficile* RT027: 4.6% in Bavaria in 2000–2009 [24], below 10% in 2011–2012 in three different regions of Germany, and 27% in Hesse [19]. However, since data from the north-eastern part of Germany are missing, a comparison with earlier data is impossible. It cannot be excluded that the molecular epidemiology of *C. difficile* in the considered region differs from other regions in the country.

Brandenburg, the federal state where the hospital in this study is located, is adjacent to the Polish border. A multicentre prospective study in Poland reported that RT027 is the most common *C. difficile* RT, accounting for 48% of clinical isolates [25]. Further studies are therefore needed to monitor the molecular epidemiology of *C. difficile* strains in the respective region and other areas in Germany over time. Another hypothesis is that *C. difficile* RT027 is a dominant strain in the study hospital and might be acquired during the stay in the hospital. It is known that epidemic strains of *C. difficile*, especially RT027, may be transmitted within hospitals and/or other healthcare facilities [26,27].

The RT027 isolates collected in this study were subjected to cgMLST to analyse their genetic relatedness in more detail. One large cluster was found, which could be divided into two subclusters. The larger subcluster A contained 25/31 (80.6%) of the RT027 isolates from the study hospital, while the second subcluster B contained the remaining 6/31 (19.4%) isolates. This finding might be explained by within-hospital transmission of a few strains. Alternatively, it is also possible that the RT027 isolates detected here represent an epidemiologically successful strain that is circulating in the region of the hospital or in Brandenburg. One reason for this is the fact that the HA-CDI cases cluster together with CA-CDI cases and the three isolates from a different hospital. To the best of our knowledge, however, there are currently no data available on the type of circulating *C. difficile* strains in Brandenburg or in the neighbouring states, including Berlin. More studies on cgMLST profiles of RT027 from different hospitals, regions, and time points are thus needed to improve our understanding of the epidemiology of RT027 strain(s) in Germany and worldwide.

Approximately 75% of CDI cases are HA-CDI, although that varies in different geographic regions [28,29,30]. However, little is known about the incidence of HA-CDI according to the ECDC definition in German hospitals, as many hospitals perform CDI surveillance according to the KISS criteria and definitions, which assess only those infections as HA that can be linked to a previous stay in the same hospital and not to healthcare facilities in general [15,16]. When the proportion of HA-CDI in this study is compared with the published ECDC data from several European countries and hospitals in 2016, it is higher than the European average (92.1% versus 74.6%) [15]. However, most hospitals included in the ECDC data are small primary-care hospitals, which mostly have a shorter length of stay of patients, less complicated cases and, consequently, a lower percentage of nosocomial infections than tertiary-hospitals [31]. This hospital is the region’s largest and only tertiary-care hospital and has a higher proportion of critically ill, multimorbid patients with multiple hospitalisations, who are likely to have a higher rate of CDI. Another possible explanation is that the data collection in our study was done with high commitment and very consistently, which may not have always been the case in other settings, considering that the case definition for HA-CDI requires an intensive study of the patient’s medical history, e.g., regarding previous hospitalisation.

In the present study, the overall mortality rate within 30 days after diagnosis was 21.4%. This is in accordance with the study by Arvand and Bettge-Weller in 2016, who reported an overall mortality rate of 19.7% in Hesse [22]. Studies conducted worldwide indicate an overall mortality rate of 8.6–18% in CDI patients within 30 days of diagnosis [32,33,34]. The all-cause mortality of 41.4% within 90 days of diagnosis of CDI in this study is remarkably higher than other published data, which report an overall mortality of 17.1–21.3% [25,32]. This might be related to the high prevalence of RT027 in this study, although we did not find a significant correlation between RT027 and mortality. However, it should be noted that the sample size was small in this study, e.g., data from the 90-day follow-up were available for only 70 patients, which might have affected the mortality rates. Therefore, the data should be interpreted cautiously.

The recurrence rates of CDI within 30 and 90 days were 15.7% and 18.6%, respectively, with an average onset after 23 days. A systematic review by Wiegand et al. revealed a CDI recurrence rate of 1–36% of cases [9]. Smits et al. reported a 15–25% recurrence rate in the first eight weeks (= 56 days) after diagnosis [8,9]. Our data are in accordance with these results.

Information regarding the implementation of additional IPC measures was available for 70 patients. Of these, 95.7% were treated with barrier nursing or single-room isolation. However, these measures were implemented in only half of CDI patients immediately after the onset of diarrhoea. A study from Canada reported that 27% of patients were isolated immediately after the onset of diarrhoea, which was highest in university and teaching hospitals (59%) [35]. Transition to sporicidal surface disinfectants occurred in 42% of CDI patients in our study, although this measure is required according to the study hospital’s internal guidance documents (hygiene plan). In the above-mentioned Canadian study, an audit of cleaning processes was performed in 72% of cases, but no information is available on a possible change of the disinfectant used for surface disinfection.

The study’s main limitation results from the relatively small number of patients, although the data were collected from a large tertiary hospital, and all CDI patients that were treated in one year were included.

## 5. Conclusions

This study shows that the prevalence of HA-CDI is rather high in a hospital in north-east Germany. We found a high recurrence rate and mortality rate within 30 and 90 days after CDI diagnosis. RT027 was the predominant *C. difficile* strain in the study hospital, whereas the prevalence of RT001 was very low. cgMLST showed the close relationship of all RT027 isolates. However, since data on the molecular population structure of this strain are currently lacking, interpretation is difficult. Our data underscores the need for rapidly initiated and specific hygiene measures to prevent and control CDI and the importance of molecular surveillance of *C. difficile* at the local and national level.

## Figures and Tables

**Figure 1 microorganisms-11-00044-f001:**
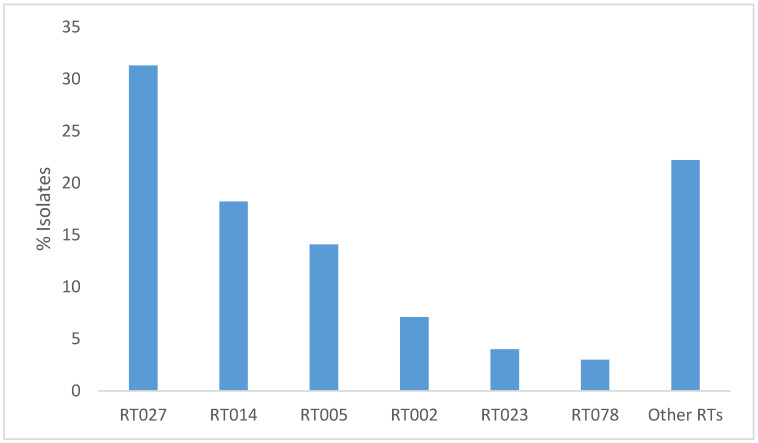
Distribution of *C. difficile* RTs among clinical isolates from hospitalised patients (n = 99).

**Figure 2 microorganisms-11-00044-f002:**
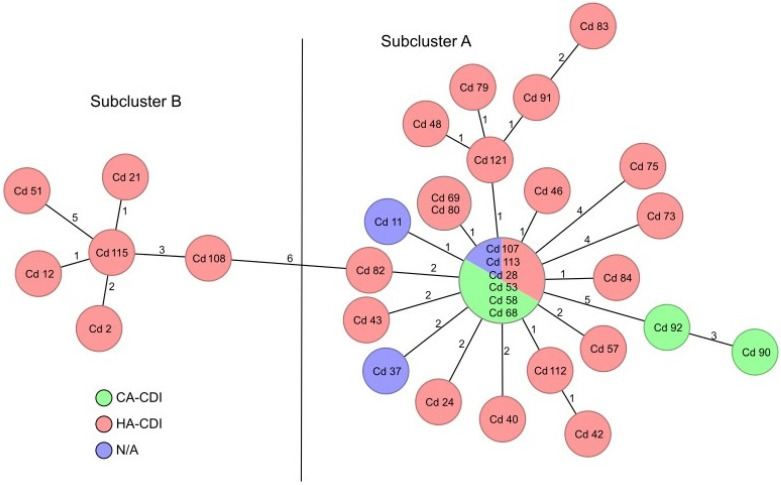
Minimum-spanning tree based on cgMLST analysis demonstrating the clonal relationship of all RT027 isolates investigated. Each circle represents the genotype based on a unique allelic profile of up to 2147 cgMLST genes (ignoring missing values in pairwise comparisons), and the numbers on connecting lines show the number of differing alleles. The circles are named by the isolate labels and sizes are related to the number of isolates with the same allelic profile. The colour of the circle provides additional epidemiological information (CA-CDI, HA-CDI, and N/A—not applicable due to the fact that the isolates Cd 11, Cd 28, and Cd 37 originated from another hospital). The black line divides the genotypes in subclusters A and B.

**Table 1 microorganisms-11-00044-t001:** Overall mortality within 30 or 90 days of diagnosis of CDI.

Parameter	Overall Mortality at:	
	30-day follow-up	90-day follow-up
Number of patients (%)	15 (21.4%)	29 (41.4%)
Time to death after diagnosis		
mean	11.6	27
median	10	21
minimum	1	1
maximum	21	70

n = 70 patients were available for 30-and 90-day follow-up.

## Data Availability

All raw reads generated were submitted to the European Nucleotide Archive (http://www.ebi.ac.uk/ena/) accessed on 18 December 2022 under accession no. PRJEB58267.

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
