# Peer review of "Molecular Epidemiology, Clinical Course, and Implementation of Specific Hygiene Measures in Hospitalised Patients with Clostridioides difficile Infection in Brandenburg, Germany"

_microorganisms, 2022, doi:10.3390/microorganisms11010044_

Round 1
Reviewer 1 Report
Overall this is a very well executed and well written study. The authors did a great job. I was struck by the 41% mortality rate at 90 days follow up after CDI. Interesting how ribotype RT027 did not have higher mortality. The article is important because it provides new sequencing data that may contradict previous studies on the epidemiology of C.diff ribotypes in Germany. I would accept with minor edits, see below.
Line 74 -Provide reference for evidence that MLST has higher discriminatory power than ribotyping.
Line 113 – Provide more details on ribotyping process. It is reference 20 the protocol by Arvand, but provide a brief description of the technique.
Reviewer 2 Report
Abstract
There is no mention of hygiene measures in the result section, but overemphasis is placed on the conclusion section.
Methods
Please describe whether this study is prospective or retrospective.
Results
What is the meaning of ‘met the inclusion criteria’ in line 128? Does it mean the ‘Screening’ or ‘Enrollment’? If that means screening, you should be mentioned about the number of enrolled CDI patients who agreed with this study. If that means enrollment, you should be mentioned about the number of screened patients and incidence should be calculated with screened CDI patients as you know. As far as I understand, this study was conducted as a prospective study, and it is almost impossible to obtain 100% agreement. Please make clear.
If possible, please present the median days of ‘hospital day of CDI onset’.
The contents of line 146-151 are recommended to be moved to line 128 for better flow.
Discussion
It is recommended to delete repeated sentences, such as line 198-202.
Line 210: Which is correct, 1000 patient-days or 10,000 patient-days?
Conclusion
Lines 300-302 also overemphasize the meaning of hygiene measures in the results of this study.
Figure 2.
Please 1. Draw the outline of each cluster 2. Mark distinctively to 3 isolates which were collected from other hospitals. 3. Mark distinctively to isolates from HA-CDI.
Is it possible to make a cluster with only one strain? Please define ‘cluster’.
Reviewer 3 Report
This paper is very interesting for me and I think will be also interesting for readers of Microorganisms. Informations from Germany often confirmed other RTs, but not 027. Now they confirmed in majority of cases causative agent of CDI RT 027, the same which was confirmed in the Southern Poland in 2017, also in other regions of Poland. Paper is interesting, however, I think for many readers will be interesting to compare C. difficile clusters from Germany with clusters of C. difficile from other countries.
Round 2
Reviewer 2 Report
Thank you for your response.